# Fitness landscape of a dynamic RNA structure

**Valerie W. C. Soo** [1,2]*, **Jacob B. Swadling**[1,2], **Andre J. Faure**[3], **Tobias Warnecke**[1,2]*

**1** Medical Research Council London Institute of Medical Sciences, London, United Kingdom, **2** Institute of Clinical Sciences, Faculty of Medicine, Imperial College London, London, United Kingdom, **3** Centre for Genomic Regulation (CRG), The Barcelona Institute of Science and Technology, Barcelona, Spain

* v.soo@lms.mrc.ac.uk (VWCS); tobias.warnecke@lms.mrc.ac.uk (TW)

**Data Availability Statement:** All sequencing data are available from the NCBI Sequence Read Archive (accession number PRJNA636762). https://www.ncbi.nlm.nih.gov/sra/?term=PRJNA636762 All other data are within the manuscript or its Supporting Information files.

## Abstract

RNA structures are dynamic. As a consequence, mutational effects can be hard to rationalize with reference to a single static native structure. We reasoned that deep mutational scanning experiments, which couple molecular function to fitness, should capture mutational effects across multiple conformational states simultaneously. Here, we provide a proof-of-principle that this is indeed the case, using the self-splicing group I intron from *Tetrahymena thermophila* as a model system. We comprehensively mutagenized two 4-bp segments of the intron. These segments first come together to form the P1 extension (P1ex) helix at the 5' splice site. Following cleavage at the 5' splice site, the two halves of the helix dissociate to allow formation of an alternative helix (P10) at the 3' splice site. Using an *in vivo* reporter system that couples splicing activity to fitness in *E. coli*, we demonstrate that fitness is driven jointly by constraints on P1ex and P10 formation. We further show that patterns of epistasis can be used to infer the presence of intramolecular pleiotropy. Using a machine learning approach that allows quantification of mutational effects in a genotype-specific manner, we demonstrate that the fitness landscape can be deconvoluted to implicate P1ex or P10 as the effective genetic background in which molecular fitness is compromised or enhanced. Our results highlight deep mutational scanning as a tool to study alternative conformational states, with the capacity to provide critical insights into the structure, evolution and evolvability of RNAs as dynamic ensembles. Our findings also suggest that, in the future, deep mutational scanning approaches might help reverse-engineer multiple alternative or successive conformations from a single fitness landscape.

## Author summary

Mutations can now be introduced into genes that code for RNAs and proteins almost at will. Yet why one mutation compromises the function of the molecule while another does not often remains unclear. This is, in part, because our main signposts for understanding the molecular basis of differential mutational effects—crystal structures–provide only very partial guidance. RNAs in particular are highly dynamic and defects can arise during multiple conformations that the RNA assumes during normal function. A single crystal structure might represent but a snapshot of all the important conformations in a large ensemble. Here we show that deep mutational scanning–a technique to generate a large

**Funding:** This work was supported by UKRI | Medical Research Council (MRC) core funding (TW, grant no. MC_A658_5TY40), a Marie Sklodowska-Curie Individual Fellowship (VWCS, grant no. 747199), and a UKRI Innovation Fellowship (JBS). This project made use of time on UK Tier 2 Joint Academic Data Science Endeavour granted via the UK High-End Computing Consortium for Biomolecular Simulation supported by the UKRI Engineering and Physical Sciences Research Council (grant no. EP/R029407/1). The funders had no role in study design, data collection and analysis, decision to publish, or preparation of the manuscript.

**Competing interests:** The authors have declared that no competing interests exist.

library of mutated versions of the original molecule–can simultaneously capture the impact of mutations that exert their effect in one of several conformations the molecule assumes during its life cycle. Deep mutational scanning can therefore be used, in principle, to study conformations that are transient or hard to observe and to better understand why and when mutations are harmful.

## Introduction

Many RNAs need to fold into defined structures to function. This includes key RNAs in information processing (e.g. rRNAs, tRNAs), RNAs with catalytic activity (ribozymes), and many smaller RNAs (e.g. microRNAs) whose biogenesis depends on base-pairing of a precursor molecule. The need to fold into specific structures and avoid erroneous intra- and intermolecular interactions constrains RNA evolution and evolvability [1,2], because at least some mutations will compromise folding, function, and fitness.

Over the last decade, mutational effects on molecular fitness have been elucidated at scale for a handful of model RNAs using deep mutational scanning experiments, both *in vitro* [3–8] and *in vivo* [9–14]. These studies have revealed complex fitness landscapes, in which both pairwise and higher-order epistasis are prevalent [12,15–17].

In some instances, mutational effects on fitness and the origins of epistasis can be rationalized with reference to a known (native) structure. It is easy to see, for example, how base-pairing in a conserved helix of a tRNA can be disrupted by a first mutation but then restored by a second mutation, leading to positive epistasis [10]. Frequently, however, the molecular foundations of variable constraint and epistasis remain obscure.

Part of the explanation for this likely rests in the fact that RNA structures are dynamic [18]. As an RNA interacts with itself and its binding partners–during biogenesis, folding, and normal function–conformational changes alter the effective genetic context of a given mutation, i.e. the context that determines mutational impact at a particular point in the life cycle of the RNA. As a consequence, a single static structure, taken as the sole representative from a dynamic conformational ensemble, can only ever act as a partial guide and will sometimes fail to inform on the contexts in which a particular mutation exerts its effects.

Deep mutational scanning experiments allow simultaneous measurement of mutational effects across multiple conformational states, however transient, as long as these states affect fitness (as measured by the experiment). The challenge is to allocate observed patterns of constraint and epistasis to these alternative conformational states, which, even if critical for function, are usually unknown and can typically not be extrapolated from knowledge of the native structure.

Here, we investigate the fitness landscape of a dynamic RNA structure that, in our assay, assumes multiple conformational states with known relevance to fitness. We consider a derivative of the group I intron from *Tetrahymena thermophila* (Fig 1A), a self-splicing ribozyme whose functional elements and key catalytic steps have been dissected in great detail using a combination of genetic, biochemical and structural approaches [19]. To measure molecular fitness and characterize epistatic interactions, we use a previously developed heterologous reporter system where the intron is embedded in a kanamycin nucleotidyltransferase (*knt*) gene (Fig 1B), placed on a plasmid and introduced into *E. coli*. This system couples self-splicing activity to fitness (Fig 1C) as intron removal is required for the reconstitution of the *knt* open reading frame, translation of which enables growth in the presence of kanamycin [20].

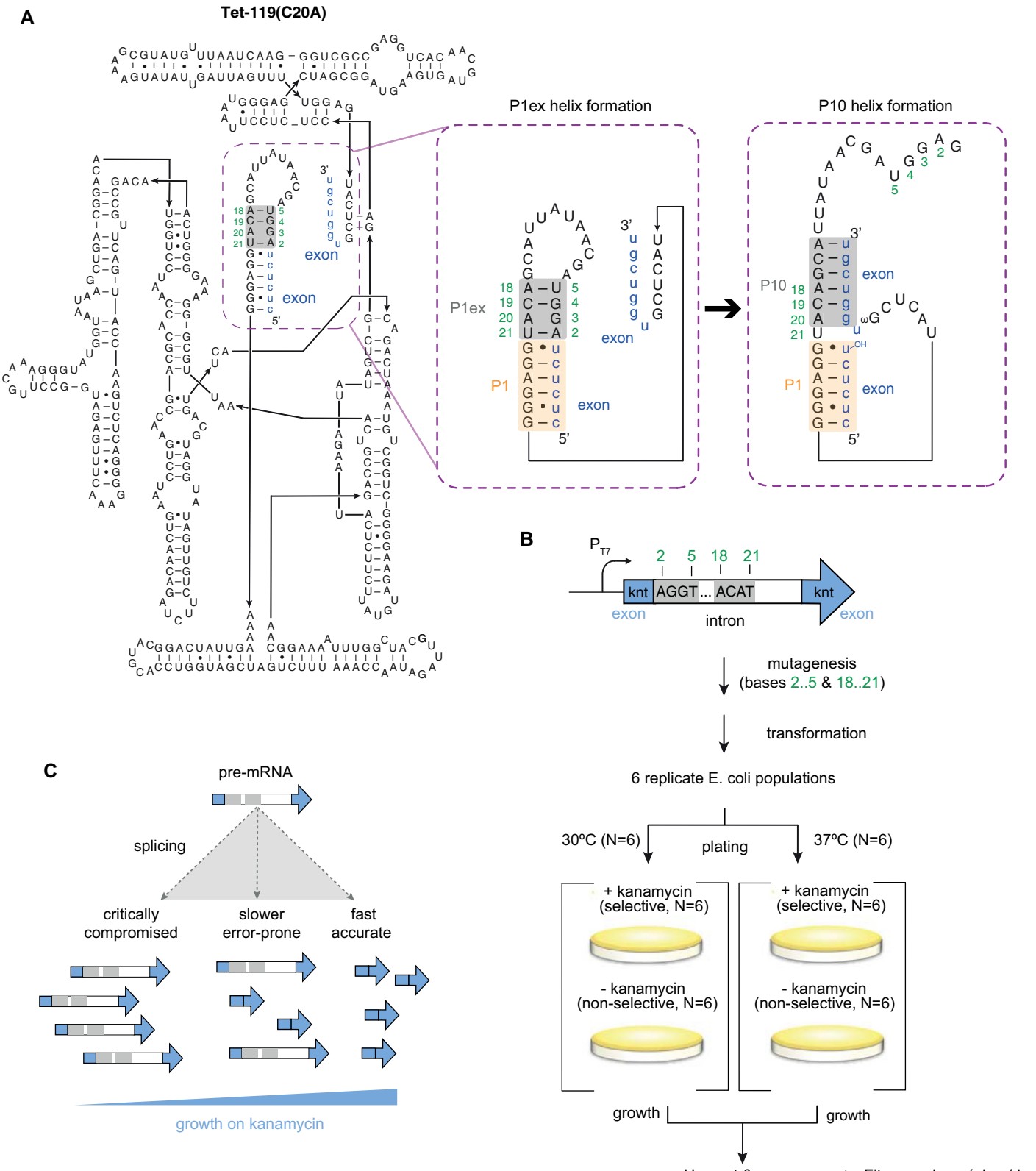

**Fig 1. Determining the fitness landscape of a dynamic RNA structure.** (A) The sequence and secondary structure of the Tet-119(C20A) group I intron with its 5' and 3' exonic context. Secondary structure conformations during sequential formation of P1ex and P10 are highlighted in the blow-ups. The two sub-regions that were

subjected to mutagenesis ($N_2..N_5$ and $N_{18}..N_{21}$) are shaded grey. (B) Schematic representation of the *knt*-intron construct, library generation, and selection protocol. (C) In the presence of kanamycin, self-splicing activity (molecular fitness) of the group I is coupled to organismal fitness as intron removal is required for reconstitution of the *knt* open reading frame.

We investigate two sub-regions in this intron, $N_2..N_5$ and $N_{18}..N_{21}$, which come together to form the P1 extension (P1ex), a 4-bp helix adjacent to the 5' splice site (Fig 1A). Importantly, following cleavage at the 5' splice site, P1ex needs to dissociate to allow formation of a second helix (P10), where one half of P1ex ($N_{18}..N_{21}$) pairs with bases at the 5' end of the 3' exon [21] (Fig 1A). Constraints on the two sub-regions are therefore asymmetric (with additional constraint on $N_{18}..N_{21}$) and pleiotropic (as $N_{18}..N_{21}$ function as part of P1ex and subsequently P10). Although the presence of neither P1ex nor P10 is strictly required for splicing [19,22,23], both helices contribute to splicing efficiency, as they facilitate splice site alignment and exon ligation and reduce non-productive alternative interactions, including the use of cryptic splice sites [21,24–27]. Mutations in P1ex and P10 have previously been shown to affect rates of catalysis at different stages of splicing [20,25,27,28], which is relevant for KNT production and, subsequently, fitness [20]. Prior work has also provided *prima facie* evidence for antagonistic pleiotropy, inferring–from a small collection of individual mutants–that overly stable pairing in P1ex might be selected against because it impedes dissociation of P1ex and therefore formation of P10 [20,25].

By measuring fitness for a large number of intron genotypes that vary at $N_2..N_5$ and $N_{18}..N_{21}$, we dissect the resulting fitness landscape to demonstrate that fitness effects of specific mutations can be allocated to distinct conformational states and used to investigate pleiotropic trade-offs. Our results provide a proof-of-principle that deep mutational scanning data simultaneously captures fitness effects arising from multiple alternative or successive conformational states. They also suggest that, in the future, this technique could be used alongside evolutionary analysis, structural modelling, and biochemical approaches to infer alternative states at scale, including those that are transient and hard to capture using traditional approaches.

## Results

We used targeted saturation mutagenesis via overlap extension PCR to generate a large library of intron variants, using a previously characterized mutant with high splicing activity [Tet-119 (C20A)] as our master sequence (Fig 1A, Materials and methods). Introns differ in the two sub-regions $N_2..N_5$ and $N_{18}..N_{21}$ but are otherwise isogenic. The library was introduced into *E. coli* and each biological replicate split into four aliquots, which were spread on agar plates that did or did not contain kanamycin and incubated at either 30˚C and 37˚C (Fig 1B, Materials and methods). After overnight incubation, genotype frequencies under selective and non-selective conditions were assayed via high-throughput amplicon sequencing (Materials and methods). This relatively short incubation time allows us to capture genotypes of intermediate fitness that would have vanished from the genotype pool in the longer term, outcompeted by a small number of genotypes with superior fitness.

Under non-selective conditions (without kanamycin, *-kan*), where production of functional KNT protein is not required for survival, our library is virtually combinatorially complete. Across 6 biological replicates and 31,269,777 sequencing reads (at 30˚C, S1 Table), we detect 65,533 of all $4^8 = 65,536$ possible genotypes (>99.99% completeness). As a consequence of the library generation protocol, and similar to prior work [3], sequences closer to the starting template are more common, increasing our power to investigate sequence space closer to the splice-competent master genotype (S1 Fig, Materials and methods).

Different genotypes with higher or lower fitness can be thought of as conceptually equivalent to different transcript species that increase or decrease in abundance. We therefore analyzed the data using a method commonly employed for counts-based differential expression analysis: DESeq2 [29]. This approach has several advantages. In particular, it is well suited to leveraging the availability of multiple biological replicates to determine significant changes in relative genotype abundance in the face of biological variability. We note that fitness estimates derived using DESeq2 are highly correlated ($r^2 = 0.91$, $P<2.2*10^{-16}$; S1 Fig, Materials and methods) to estimates from an alternative method, DiMSum [30,31], which explicitly models the main sources of variability in deep mutational scanning data.

Under selective conditions (+*kan*), colony formation is much reduced (S2 Fig) and the majority of genotypes (42193/65536 = 64%) experience a significant (at $P_{adj}<0.05$) drop in frequency, while only 6.5% (4286/65536) become significantly more common, leading to a precipitous decline in overall genotype diversity (Fig 2A and 2B). Individual P1ex genotypes previously found to exhibit increased splicing efficiency have concordant effects in our assay (S3 Fig). Is this reduction in diversity consistent across replicates, in such a way that we end up

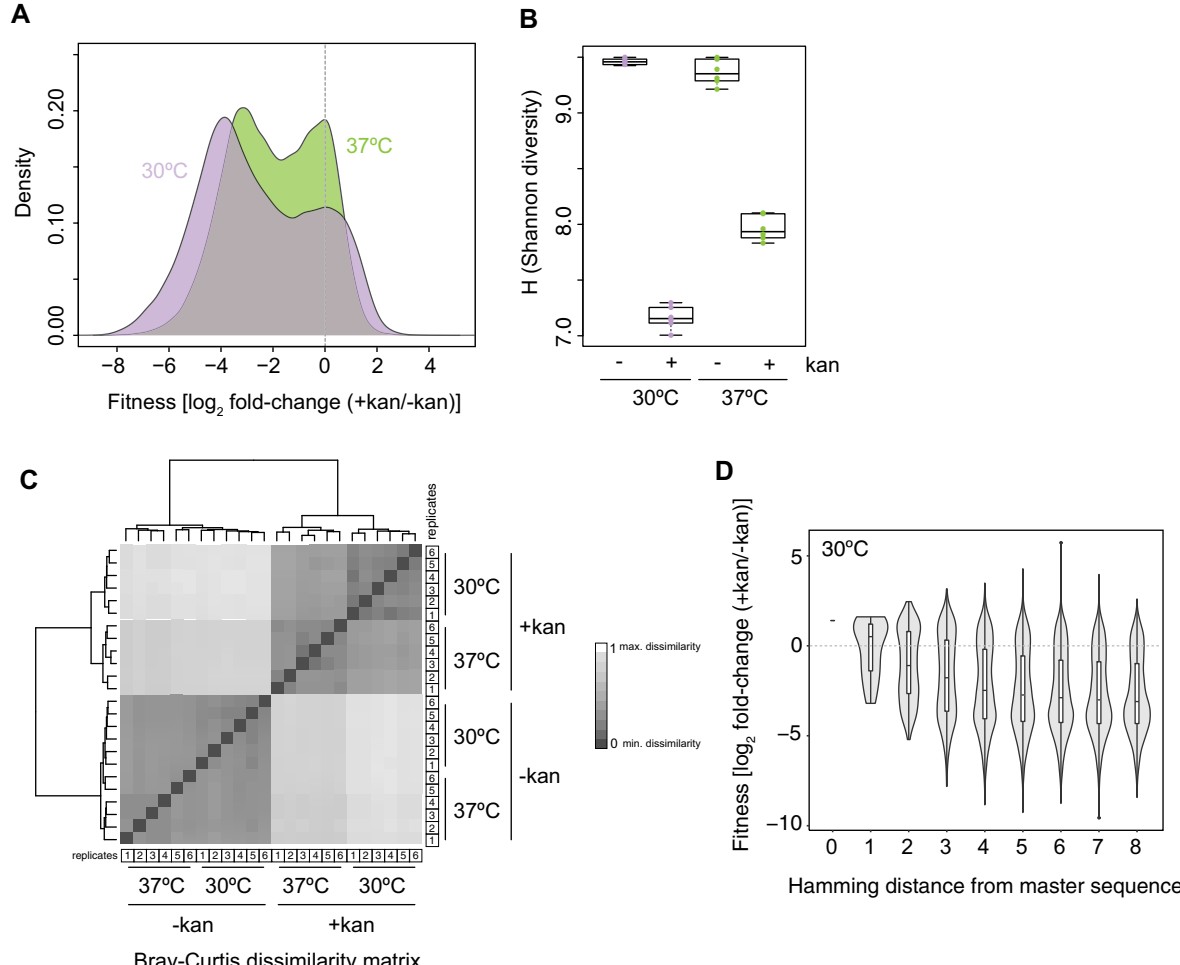

**Fig 2. Fitness across intron genotypes.** (A) Distribution of fitness effects at 30°C and 37°C. (B) Shannon diversity of intron genotype pools under different conditions. (C) Similarity in genotype pool composition across all replicates and conditions measured as Bray-Curtis (BC) dissimilarity, where BC = 1 indicates maximum dissimilarity between samples. (D) Fitness of intron genotypes at 30°C as a function of Hamming distance (i.e. the number of mutational steps away from the master sequence).

with similarly altered genotype pools? To answer this question we computed Bray-Curtis dissimilarities, a metric we adopt from the ecology literature. Bray-Curtis dissimilarity captures both the number of species in an ecosystem and their relative abundance to provide an integrated measure of ecosystem diversity. Using this metric, we find that the genotype pools from different replicates are more similar within a given condition (±*kan*, 30/37˚C) than between conditions (Fig 2C), indicating consistent changes to genotype diversity following exposure to kanamycin.

Similar to the fitness landscapes of other RNAs and proteins [32], the distribution of fitness effects across genotypes is bimodal and average fitness decreases as the number of mutations away from the master sequence (= Hamming distance) increases (Fig 2A and 2D).

## Fitness effects across mutant genotypes support selection against excess stability in P1ex

Prior work on both tRNA and snoRNA found fitness defects to be more pronounced at 37˚C compared to 30˚C [11,14], consistent with destabilization of folded structures as a key determinant of mutant fitness. We observe the opposite (Fig 2A). While fitness estimates for individual genotypes are highly correlated between 30˚C and 37˚C (S4 Fig, $\rho = 0.75$, $P < 2.2*10^{-16}$), fitness impacts are quantitatively milder, on average, at the higher temperature. This is in line with the suggestion that excess stability of P1ex secondary structure compromises efficient splicing [20], as kinetic traps should, on average, be easier to escape and misfolding issues be less severe at 37˚C. In support of this explanation, we find greater predicted stability of the intron and higher GC content to be associated with larger decreases in fitness (Fig 3A and 3B; Materials and methods). At the same time, genotypes that cannot form *any* on-target base-pairs also exhibit low fitness (0 strong/weak base-pairs in Fig 3C). In contrast, genotypes where helices *are* formed, but the constituent base-pairs are weak (A-U), as found in the *T. thermophila* native structure (S5 Fig), typically do well (Fig 3C).

The need to avoid an overly stable P1ex helix is further evident when looking at patterns of epistasis. In contrast to most other RNA deep mutational scanning studies [16], we observe an enrichment for positive rather than negative pairwise epistasis when considering single and double mutations away from the master sequence (Fig 3D). In some instances, positive epistasis corresponds to the classic case where a base-pair is broken by each of two individual mutations but restored when these mutations are combined. However, we observe multiple cases of strong positive epistasis that do not conform to this model. Notably, many such cases involve A20C and G3U (Fig 3E), the only two mutations capable of generating a helix with four paired bases. Any further mutation elsewhere in the two sub-regions will abolish perfect complementarity in P1ex. Almost always, the reduction in fitness upon adding this second mutation is less severe than expected under an additive model of mutational effects, in line with selection against excess stability. This highlights that positive epistasis can result not only from selection to maintain base pairing but also from selection to prevent it.

## Machine learning facilitates allocation of mutational effects to distinct conformational states

Although simple metrics like stability and GC content are related to fitness, they are overall poorly predictive (GC content: $\rho = -0.10$; predicted free energy: $\rho = 0.17$; Fig 3A and 3B), suggesting a more complex landscape of constraint than one exclusively defined by a P1ex structural stability threshold. To better understand how specific mutations affect fitness and whether they do so in a P1ex and/or P10 context, we sought to determine the contribution of individual nucleotides to fitness systematically. To this end, we trained extreme gradient

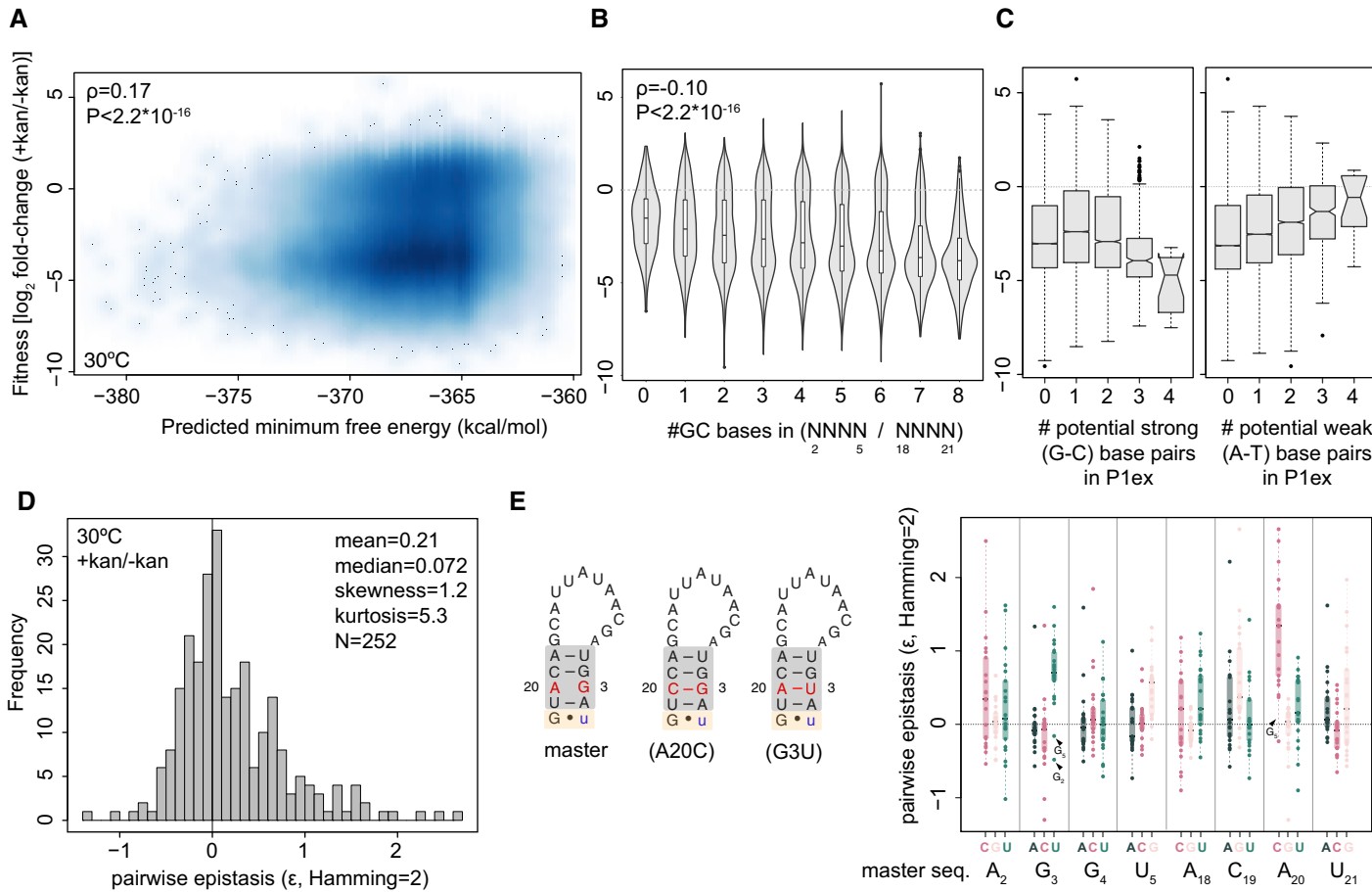

**Fig 3. Causes and correlates of variable fitness across intron genotypes.** (A) Fitness weakly correlates with predicted minimum free energy of the intron. For orientation, note that the predicted minimum free energy (ΔG) of the master sequence is -362.8 (B) Fitness varies according to the number of guanosines or cytosines (#GC) in the $N_2..N_5$ and $N_{18}..N_{21}$ regions. (C) Fitness varies as a function of the number of strong or weak base-pairs that could be formed in P1ex assuming that base-pairing follows the established master/wildtype pattern (see Fig 1A). (D) Distribution of pairwise epistasis values for genotypes that are two mutations away from the master sequence (Hamming distance = 2). ε>0 indicates positive epistasis, ε<0 indicates negative epistasis. (E) Pairwise epistasis for genotypes in (D) by position and mutation. Diagrams on the left highlight the $N_3/N_{20}$ couple, where mutations that are predicted to lead to base-pairing are associated with positive epistasis.

boosted decision tree (XGboost) models [33] to predict fold-changes (+*kan* vs. -*kan*) solely from nucleotide identities at $N_2..N_5/N_{18}..N_{21}$. For both 30˚C and 37˚C, we find that fold-changes predicted from the models are well correlated with observations (30˚C ρ = [0.63, 0.84], P<2.2x10⁻¹⁶; 37˚C ρ = [0.63, 0.83], P<2.2x10⁻¹⁶, see Materials and methods for calculation of correlation ranges). We estimate that these models account for ~80% of the explainable genetic variance. Providing additional RNA-wide properties as features for prediction (e.g. RNAfold-predicted stability or ensemble diversity) does not improve model performance (S2 Table), suggesting that the models capture key emergent properties from the underlying primary sequence. In addition, confining analysis to genotypes whose change in relative abundance was judged significant by differential abundance analysis, does not improve prediction accuracy. In fact, prediction accuracy is higher when these genotypes are included (S2 Table). This suggests that there is latent information in the differential abundance of low-abundance genotypes that can be leveraged by our machine learning approach to improve prediction accuracy.

The contribution of individual features to prediction accuracy can be assessed globally by considering the gain in classification accuracy when a leaf in the tree is split according to that

feature. However, computing such *gains* does not provide directionality of effect nor the ability to assess contribution locally, i.e. for individual genotypes. We therefore additionally computed Shapley additive explanation (SHAP) values [34,35], which provide a framework for interpreting the impact of individual features on model prediction in a machine learning context, and contain information about both sign and magnitude of the contribution.

In our case, a feature corresponds to having or not having a particular nucleotide (e.g. cytosine) at a given site (e.g. $N_{21}$). In some instances (e.g. $G_5$, Fig 4A), nucleotide identity affects fold-change prediction consistently in the same direction across genotypes, although the precise contribution might vary from genotype to genotype (equivalent to magnitude rather than sign epistasis). In other cases (e.g. $U_3$, Fig 4A), the identity of a nucleotide at a particular site only substantively contributes to predictions for a small number of genetic backgrounds.

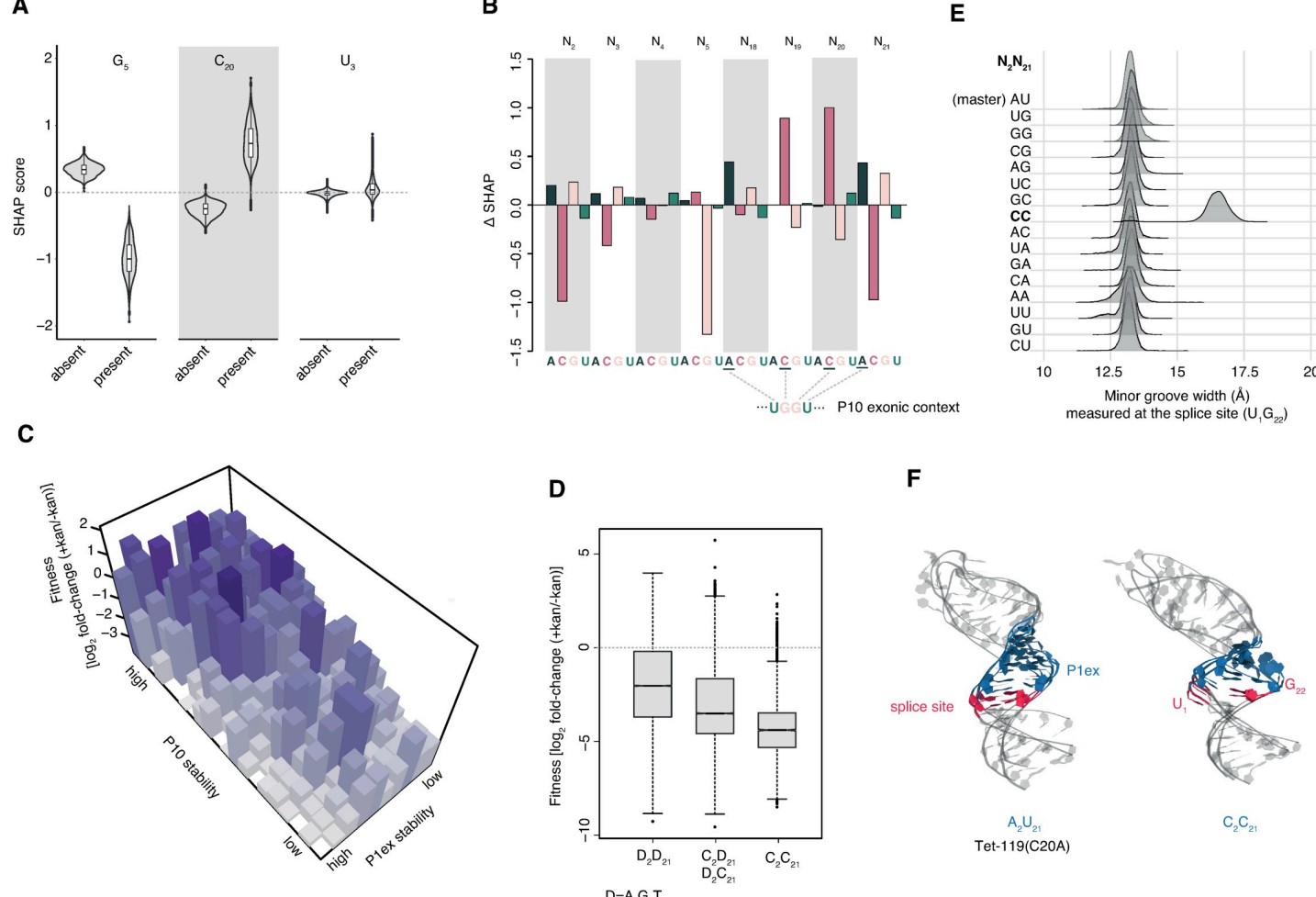

**Fig 4. Assessing the contribution of individual nucleotide identities to fitness across multiple structural conformations.** (A) Contribution to XGBoost-predicted relative fitness across all intron genotypes, as measured by Shapley's additive explanation (SHAP) scores, of three example site/nucleotide features. More positive SHAP scores are associated with higher fitness. (B) The average contribution across all genotypes of all individual site/nucleotide features, measured as $\Delta SHAP = SHAP_{present} - SHAP_{absent}$, where $SHAP_{present}$ and $SHAP_{absent}$ correspond to the mean SHAP score of all genotypes where a given nucleotide at a given site is present and absent, respectively. (C) Fitness landscape at 30°C as a function of RNA stability of P1ex and P10 across all genotypes assuming bases are aligned to pair as in the master/wildtype structure (see Fig 1A, Materials and methods). There are 211 unique energy values across all $4^8$ P1ex genotypes. These were consolidated into ten bins of increasing stability for visualization purposes. The 21 unique energy values across $4^4$ P10 genotypes are shown in full as 21 bins of increasing stability. Bar heights correspond to the median fitness in each bin. (D) Fitness as a function of $N_2/N_{21}$ genotype, with a focus on cytosines. (E) Minor groove width associated with different $N_2/N_{21}$ genotypes as determined using molecular dynamics simulations (see Materials and methods). (F) Three overlaid representative conformations of the P1/P1ex helix (randomly sampled from the final 50 ns of each simulation) for the master sequence and the $C_2/C_{21}$ genotype.

Fig 4B summarizes the average contribution of each site/nucleotide feature to the prediction by computing ΔSHAP, defined here as the mean SHAP value across genotypes where a given nucleotide at a given position is present minus the mean SHAP value across genotypes where the nucleotide at the same position is absent. Notably, the strongest positive contributions involve nucleotides that allow on-target base-pairing during formation of P10 ($A_{18}$, $C_{19}$, $C_{20}$, $A_{21}$, Fig 4B). This suggests that, even though not essential for splicing [23], P10 pairing is a major driver of differential fitness in our system. In contrast, there are no strong positive contributions from the nucleotides exclusive involved in P1ex ($N_2..N_5$). This supports earlier models, which argued that P1ex function is largely independent of sequence as long as minimal structural requirements such as avoidance of excess stability are satisfied [27,28,36]. Rather, $N_2..N_5$ is principally governed by negative constraints, where the presence of specific nucleotides is associated with decreased fitness (Fig 4B). That negative constraints (on P1ex) and positive constraints (on P10) jointly govern fitness is perhaps most clearly evident when fitness is displayed as a function of P1ex and P10 helical stabilities across genotypes (Fig 4C, see Materials and methods).

One specific negative constraint involves bases $N_2$ and $N_{21}$, where the presence of cytosines is associated with a strong negative contribution to fitness (Figs 4B and S6). This observation is consistent with prior experiments in the wildtype P1/P1ex context (S5 Fig) where an 80% (40%) decline in splicing activity was observed when $A_2$-$U_{21}$ was replaced with $G_2$-$C_{21}$ ($C_2$-$G_{21}$) [28]. We find fitness defects to be particularly pronounced when cytosines are present at both these sites ($C_2/C_{21}$, Fig 4D). In the master and wild-type *T. thermophila* sequence, $N_2$ and $N_{21}$ form a base-pair directly adjacent to the splice site $U_1$-$G_{22}$ (Figs 1A and 3E). We therefore suspected that cytosines at these positions might disturb splice site geometry. To investigate this further, we carried out molecular dynamics simulations (see Materials and methods) of all 16 possible $N_2/N_{21}$ combinations in an otherwise isogenic Tet-119(C20A) context. Considering a catalogue of features [37] that describe base-pairing geometry (stagger, roll, twist, etc. see Materials and methods) we find that $C_2/C_{21}$ –uniquely–leads to a radical structural deformation of minor groove geometry (Figs 4E, 4F and S7 and S1 Movie), as the splice site $U_1$ rotates out of the helix core and $G_{22}$ mis-pairs with $C_2$. This likely disturbs splice site definition and key tertiary contacts between the P1 substrate and the catalytic core of the ribozyme [38–41], consistent with poor splicing.

Finally, $G_5$ makes a strong negative contribution to fitness, both on average and across genotypes (Fig 4A and 4B). It is interesting to note in this regard that in many naturally occurring introns, including the native *T. thermophila* intron (S5 Fig), no pairing is observed at $N_5$-$N_{18}$ resulting in a P1ex helix that is only three bases long. This suggests that having a base-pair at this position and/or extending the helix beyond three bases often interferes with efficient splicing (S6 Fig). However, unlike in the case of $N_2$-$N_{21}$, the negative contribution of $G_5$ is not mirrored on the other side of the helix (at $N_{18}$); we therefore believe that $G_5$ might have negative fitness consequences outside the P1ex context that remain to be deciphered.

## Asymmetric fitness effects allow inference of pleiotropy

Given its role in participating in both P1ex and P10, $N_{18}..N_{21}$ has to satisfy an additional layer of constraint and mutations at $N_{18}..N_{21}$ are expected to be pleiotropic. We asked whether such additional constraint may be reflected in the relative contributions that different site/nucleotide features in $N_2..N_5$ versus $N_{18}..N_{21}$ make to predictions. We find this to be the case: a significantly larger proportion of gains in the model is attributable to $N_{18}..N_{21}$ (Fig 5A). This asymmetry is also reflected in patterns of epistasis. When we consider pairwise interactions within $N_2..N_5$ (with $N_{18}..N_{21}$ fixed as ACAU), within $N_{18}..N_{21}$ (with $N_2..N_5$ fixed as AGGU) or

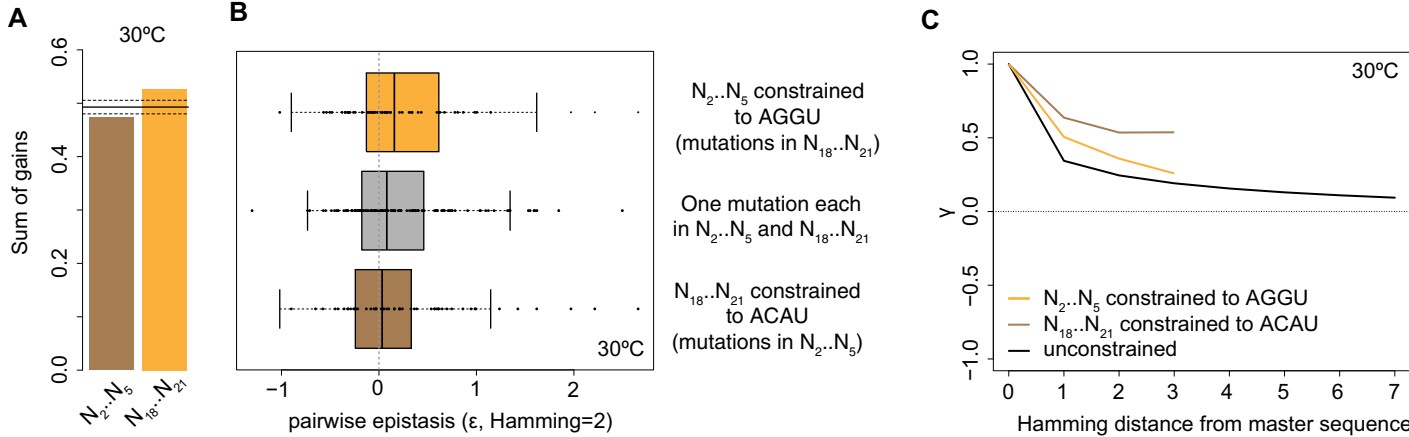

**Fig 5. Asymmetric fitness effects across the $N_2..N_5$ and $N_{18}..N_{21}$ sub-regions.** (A) Proportion of gains in the model (see main text) contributed by site/nucleotide identity features at $N_2..N_5$ and $N_{18}..N_{21}$. The solid line corresponds to the mean contribution made by a sub-region across 100 random samples, where individual gains are randomly shuffled across site/nucleotide identity features. Dashed lines correspond to 95% confidence intervals. (B) Pairwise epistasis for double mutants where both mutations are located in $N_{18}..N_{21}$ (orange), both mutations are located in $N_2..N_5$ (brown), or $N_2..N_5$ and $N_{18}..N_{21}$ carry one mutation each (grey). (C) The correlation of fitness effects ($\gamma$) of intron mutants at various mutational distances from the master sequence.

across helices (with one mutation each in $N_2..N_5$ and $N_{18}..N_{21}$), we find a tendency for positive epistasis to be more prevalent within $N_{18}..N_{21}$ than cross-helix and particularly compared to $N_2..N_5$ (Fig 5B, Wilcoxon text, P<0.1) Thus, positive epistasis is more common, on average, for mutations at nucleotides $N_{18}..N_{21}$, consistent with pleiotropic constraint. Distinct landscapes of epistasis in $N_2..N_5$ versus $N_{18}..N_{21}$ are also evident when we consider higher-order epistasis by computing the correlation of fitness effects ($\gamma$) [42] at different Hamming distances from the master sequence. Finally, to further illustrate asymmetric fitness effects across the P1ex helical divide, we carried out a simple mirror test, where we compare the fitness of a given genotype (e.g. $A_2AAG_5/C_{18}TTT_{21}$) to its mirror image across the helix axis (here $T_2TTC_5/G_{18}AAA_{21}$). To provide a fair comparison, we only considered genotypes and their mirror genotypes that are at equal Hamming distance (d = 2) from the master sequence. In line with strongly asymmetric fitness effects motifs, we find only a weak, non-significant correlation between the fitness of mirrored genotypes ($\rho = 0.21$, P = 0.4; N = 19). These results serve as a reminder that, even though restoration (e.g. flipping a G-C to a C-G base-pair) is commonly used to demonstrate the importance of base-pairing and helix formation, two sides of any given helix need not necessarily be equivalent. In fact, for RNAs in general we expect asymmetry to be common, caused by differential involvement in folding intermediates and alternative conformational states, but also specific modifications and interactions with chaperones and other proteins and RNAs. Asymmetric effects are likely prevalent even in helices where base-pairing is of pre-eminent concern. tRNAs, for example, are post-transcriptionally modified and interact with proteins (e.g. tRNA synthetases) in a highly asymmetric manner.

## Signatures of asymmetric constraint during the evolution of Tetrahymena P1ex?

Can we detect signatures of asymmetric constraint and pleiotropy in the evolutionary history of P1ex/P10? To find out, we considered the distribution of variants/substitutions across orthologous introns in different Tetrahymena strains/species. We used BLAST to identify 56 homologous Tetrahymena introns and generated an alignment of these sequences with the aim to determine whether nucleotides $N_2-N_5$ are subject to different constraints than $N_{18}-N_{21}$,

mirroring our experimental findings. We find that, while there is variation in the intervening loop, both $N_2$-$N_5$ and $N_{18}$-$N_{21}$ are perfectly invariant (S5 Fig). There is therefore, unfortunately, insufficient genetic heterogeneity in this clade to contrast patterns of evolution and experimental results directly, beyond lending support to the notion that P1ex/P10 formation and composition appear functionally important. Note here that analysing P1ex evolution beyond Tetrahymena is problematic: the intron is absent from close relatives of Tetrahymena [43] and distant relatives have little similarity in terms of P1-proximal architecture and exonic context. We therefore think that aligning and comparing distant orthologs has limited merit.

Many self-splicing introns have a chequered history involving frequent loss, gain, and horizontal transfer, which complicates tracking substitutions in a phylogenetic context [44,45]. Other RNAs might therefore prove more amenable to the study of pleiotropy and asymmetric constraint. In particular, we think that riboswitches would make an excellent subject for further study. First, riboswitch function involves the formation of competing helices (typically including participation of some but not all nucleotides in more than one helix). Second, riboswitches are common and more stably inherited than self-splicing introns, facilitating evolutionary and comparative analysis. Third, riboswitches are relatively small and can therefore be mutagenized systematically [46]. Finally, riboswitches can either be hooked up to a reporter gene or the activity of (metabolic) downstream genes themselves can be measured providing a means to map genotype to fitness.

## Discussion

Our study provides a proof-of-principle that deep mutational scanning experiments can capture multiple fitness-relevant conformational states simultaneously, providing a window onto the fitness of RNAs in their true ensemble state. The capacity to capture multiple structural states in a one-pot experiment brings both opportunities and challenges. Challenges, because mutant fitness need not be interpretable in context of single (native) structure. In fact, mapping fitness effects onto a single native structure might prove misleading at sites where a dominant contribution to fitness comes from non-native, alternative, or transient conformations or where mutational effects are pleiotropic. At the same time, capturing ensembles brings opportunities: data from deep mutational scanning experiments might help us identify residues whose contribution to fitness is large but not easily explained when considering the native structure and prioritize these residues for follow-up studies. In the context of our study, $G_5$ stands out as a residue that deserves further investigation, its significant contribution to fitness poorly rationalized by the current stability model.

Our study does not aim to provide a detailed dissection of fitness defects for individual genotypes. Splicing might be compromised for a number of mechanistically distinct reasons; some related, some unrelated to the need to successively form P1ex and P10. Some variants might lead to kinetic problems (e.g. slow dissociation of P1ex), others might trigger misfolding of P1 or increase reverse splicing. Yet others might inadvertently promote the use of cryptic splice sites, as documented previously [47], or lead to undesired interactions with other RNAs or proteins in *trans*. Instead of dissecting the mechanistic basis of individual instances of splicing failure, we have leveraged fitness data across genotypes to allocate fitness effects to one of two alternative RNA conformations, which had previously been identified by painstaking biochemical dissection. Would we have been able to predict the existence of these two structures from the data *de novo*? And would we be able to do so for other RNA structures, including for RNAs where the true number of fitness-relevant alternative/successive conformations is unknown? The short answer to the first question is likely to be no, although we do not show this formally here. Our mutagenesis strategy was not geared towards blind *de novo* prediction

but focused on establishing a proof-of-principle that multiple conformational states leave a joint mark on the fitness landscape. We therefore only targeted a small portion of the molecule within which interactions can take place. Without prior knowledge or constraints, the conformational search space would span the entire *knt*-intron construct, which is large and allows for many potential interactions. Having a high-resolution genotype-fitness map for the entire RNA will increase the chances of inferring specific structures *de novo*. In addition, bounding the search space, for example by assuming–as one might for riboswitches–that alternative conformations are formed locally, should make *de novo* prediction from mutational scanning experiments considerably easier.

While our data are not suitable for *de novo* structure prediction, Schmiedel and Lehner recently demonstrated that deep mutational scanning data *can* be used for just this purpose. Exploiting covariance in fitness between particular residues as inputs for constraint-based modelling of physical interactions, the authors managed to reconstruct secondary and tertiary protein structures with high accuracy [48]. In principle, constraint-based modelling could be used in a similar manner to reconstruct RNA structures. The general approach here is analogous to using covariation of substitutions in multiple sequence alignments, which has underpinned recent advances in protein fold prediction [49,50]. However, we believe that deep mutational scanning data will be most powerful as part of an integrated approach to structure determination, deployed alongside analysis of evolutionary covariance patterns, molecular dynamics simulations, and tools to probe and predict RNA structure. When used as part of such a wider complementary toolkit deep mutational scanning experiments might, ultimately, help us to reverse-engineer dynamic interactions and critical non-native states from a single fitness landscape and provide a better, ensemble-based understanding of RNA evolution and evolvability.

## Materials and methods

### Construction of mutant intron library

The plasmid backbone of Tet-119 is derived from *E. coli-Thermus thermophilus* shuttle vector pUC19EKF-Tsp3 [51], which contains a ColE1 *ori*, an ampicillin resistance marker gene, and the *knt*-intron sequence under the control of a *slpA* promoter [20]. The *knt*-intron construct was made previously by inserting the intron at nucleotide 119 downstream of the translational start site of *knt*. To maintain base-pairing with the 3' exon to form P10 and so as not to introduce amino acid substitutions into KNT, nucleotides 15–20 were altered from 5'-TACCTT-3' (in the wild-type T. *thermophila* intron variant) to 5'-ACGACC-3'. Due to the change in nucleotides 19–20 from 5'-TT-3' to 5'-CC-3', nucleotides 3–4 were altered from 5'-AA-3' to 5'-GG-3' to maintain base-pairing within the P1ex region. However, *E. coli* strains bearing this intron variant were not viable when challenged with kanamycin, indicative of insufficient splicing activity [20]. Tet-119(C20A) was subsequently identified in a screen for mutants that rescued the splicing defect [20].

Upon receipt of Tet-119(C20A), a gift from Feng Guo (UCLA), we amplified the entire *knt*-intron sequence (using primers knt-rz-f and knt-rz-r, S3 Table) and subcloned it into the NdeI/XhoI sites of a pET-22b(+) plasmid (Merck Millipore) so that its expression is driven by an IPTG-inducible T7 promoter. To make the mutant library, all eight nucleotides in the two sub-regions were mutated to all possible nucleotides ($4^8$ variants) using overlap extension PCR coupled with oligonucleotides containing mixed bases at these sites (S8 Fig and S3 Table). Note that this procedure, in contrast to protocols employing doped oligonucleotides, will preferentially amplify sequences closer to the starting template as oligos closer to the starting template will bind the template better during PCR.

Oligonucleotides were from Integrated DNA Technologies, and all PCRs were carried out using Q5 High-Fidelity DNA polymerase (New England Biolabs). All DNA fragments were purified from agarose gel (Monarch DNA Gel Extraction kit, New England Biolabs) to reduce carry-over of residual contaminants.

The mutated pool of introns was then ligated into pET-22b(+), and the ligated products were electroporated into competent *E. coli* DH5a (New England Biolabs) cells according to standard procedures [52]. After electroporation, cells were recovered in SOC medium at 37˚C for 1 hour. Recovered cells were then grown on LB agar containing 100 μg/mL carbenicillin at 37˚C for 16 hours. The next day, the total number of transformed colonies was estimated to be ~5.5 x $10^5$, corresponding to at least 8-fold oversampling of the target library size of $4^8$ variants. All transformed colonies were scraped off the agar plates and pooled in 10 mL LB + 100 μg/mL carbenicillin. Half of the pooled cells were archived at -80˚C, and the remaining half was harvested for plasmid extraction (QIAprep Spin Miniprep).

## Growth under selective and non-selective conditions

The extracted plasmids from the mutant library were re-electroporated into *E. coli* BL21(DE3) as previously described. For each transformation, 13 fmol of the plasmid library (corresponding to 59 ng) was mixed with 100 μL of electrocompetent bacterial suspension. After electroporation, cells were recovered in SOC medium at 37˚C for 1 hour prior to a brief centrifugation (2,500xg, 5 min). The supernatant was removed, and the cells were washed gently with LB. After resuspending the washed cells in 0.5 mL LB, half of the suspended cells (0.25 mL) were used for experiments at 37˚C, the other half for experiments at 30˚C. For each temperature, a 125-μL aliquot was spread on an LB agar containing 25 μg/mL kanamycin, while another 125-μL aliquot was spread on an LB agar without kanamycin. Other supplements in both media, were 100 μg/mL carbenicillin, 50 μM IPTG and 0.2% rhamnose. Agar plates were then incubated overnight at either 37˚C or 30˚C. A total of six replicate transformations was carried out, but with only two replicate transformations being conducted on the same day. After incubation, colonies that formed on the agar plates with or without kanamycin were scraped off and pooled using 3 mL LB containing 100 μg/mL carbenicillin. A 1 mL aliquot of the pooled bacterial suspension was used for plasmid extraction (QIAprep Spin Miniprep) whereas the remaining pooled aliquot was archived at -80˚C.

Note here that the relatively short incubation time (overnight), along with deep sequencing coverage and the presence of multiple biological replicates allows us to assess, in a statistically robust manner, the performance of genotypes with intermediate fitness. If we had measured after several days of culture, genotypes with greater relative fitness would have spread further through the population, at the cost of less fit genotypes, many of which would likely have been eliminated. We kept exposure relatively short so that we could see a clear differential response to kanamycin while still being able to monitor more than just a handful of the very fittest genotypes.

## Library preparation and sequencing

An aliquot (3 fmol each) of the plasmids extracted from the selected and non-selected populations was used for PCR (24 cycles) to amplify a 204-bp sequence spanning the P1ex region using a pair of adapter-linked primers (C20Aseq-f and C20Aseq-r, S3 Table). The resulting amplicons from each replicate/strain were cleaned up using the Monarch PCR & DNA Cleanup kit (New England Biolabs). Next, Illumina indices (Nextera XT dual indexing) were incorporated into the adapter-linked amplicons in a second round of PCR (8 cycles), and the resulting index+adapter-linked amplicons were purified using Ampure XP beads. Index incorporation was confirmed with Agilent Bioanalyser HS-DNA. After quantifying the DNA

concentration of the index+adaptor-linked amplicons using Qubit (High Sensitivity DNA Assay), each was normalized to 2.5 nM and then combined to make an equimolar pool. The amplicon pools were subjected to 100-bp paired-end sequencing on an Illumina HiSeq 2500 v4 sequencer. To guard against batch effects, we sequenced samples following a balanced design where each of the 24 samples (6 replicates x 2 temperatures x 2 conditions), along with samples from other conditions not described in this manuscript, was split into three, and one third each allocated to one of three HiSeq lanes for sequencing. Split samples cluster tightly together on PCA, suggesting that batch effects are negligible. Raw reads have been deposited in the NCBI Sequence Read Archive under accession PRJNA636762. Read/genotype counts after filtering (see below) are provided in S1 Table.

### Read processing and fitness estimates

We quality-filtered reads and estimated fitness using two different pipelines. In the first pipeline, we treated the data as one would when conducting a differential expression experiment, where individual genotypes correspond to individual RNA species in a complex pool of transcripts. Reads were trimmed using Trimmomatic v 0.35 (HEADCROP:5 MINLEN:95) and subsequently filtered for base quality $\geq 30$ at the bases targeted by mutagenesis. Imposing stringent quality cut-offs across the untargeted backbone does not affect results, leads to the removal of many more reads and is needlessly conservative since most deviation here should be owing to sequencing errors. The relative fitness of each genotype (along with adjusted significance values, $P_{adj}$) was then estimated using DESeq2 (implemented in R) as a log2-fold change in abundance of a given genotype in six replicates treated with kanamycin compared to six replicates without kanamycin.

For comparison, fitness estimates were computed with DiMSum v0.3.2.9000 (https://github.com/lehner-lab/DiMSum) [30,31], which derives final fitness estimates as an error-weighted sum of replicate fitness values, after computing wildtype-normalized fold changes at the replicate level. DiMSum was run with the following parameters: *cutadapt5First*: GGGGAT GATGTTAAGGCTATTGGTGTTTATGGCTCTCT, *cutadapt5Second*: CGGTCTTGCCTTT TAAACCGATGCAATCTATTGGTTTAAAGACTAGCTACCAGTGCATGCCTGATAACT TTTCCCTCC, *cutadaptCut3Second*: 1, *cutadaptMinLength*: 20, *cutadaptErrorRate*: 0.2, *usearchMinlen*: 20, *wildtypeSequence*: AGGTagcaatattacgACAT, *maxSubstitutions*: 8.

As highlighted above, fitness estimates are highly concordant between the two pipelines (S1 Fig). Fitness estimates for all genotypes from both methods are provided in S1 Table.

### Computation of summary measures

Shannon diversity and Bray-Curtis dissimilarity were calculated using the *diversity* (index ="Shannon") and *vegdist* (method ="bray") functions from the R package *vegan*. Skewness and kurtosis were calculated using the *skewness* and *kurtosis* functions from the R package *moments*. To allow direct comparison to prior results [16], pairwise epistasis was calculated as $\varepsilon = \log_{10}(f_{master} {}^* f_{m1,2} / f_{m1} {}^* f_{m2})$, where $f_{master}$ is the fitness of the master sequence and $f_{m1}, f_{m2}$, and $f_{m1,2}$, are the fitness values of the two single-nucleotide mutants and the double mutant, respectively, as calculated by the DiMSum pipeline. Note that fitness in this pipeline is evaluated relative to the master sequence whose fitness is set to 1.

### Computation of RNA structural features

Minimum free energies (MFE) of the different intron genotypes was computed using RNAfold from the Vienna package (v2.4.3, −−noPS -p -d2—MEA -T 37/30), using the intron with ±10

flanking nucleotides, which is sufficient for splicing [53]. Results are qualitatively identical when we consider the intron along with the entire *knt* open reading frame instead.

## Machine learning

Extreme gradient boosted (XGBoost) decision trees were implemented using the *xgboost* and *caret* packages in R, with nucleotide identities encoded via one-hot encoding. Two-thirds of the genotypes were used for training and one third for testing, with 5-fold cross-validation. Hyperparameters were tuned via grid search [nrounds = c(100, 200, 500, 1000), eta = c (0.01,0.05,0.1, 0.3), max_depth = c(4,6,8, 10), subsample = c(0.5, 0.75, 1.0), min_child_weight = c(5, 10, 20)]. Two parameters, colsample_bytree and gamma were set to 1. Models were then built using xgbTree using the RMSE metric to minimize (method = "xgbTree", objective = "reg:linear", metric = "RMSE"). Predictions are based on the best parameters after tuning. We also carried out equivalent training for subsets of the data significant at the $P_{adj} = 0.05$ or (further restricted) $P_{adj} = 0.01$ level as well as on the Wald statistic provided by DESeq2 instead of log2-fold changes (S2 Table). However, we found no improved or worse performance in prediction accuracy when using the Wald statistic or censored sets of genotypes. As highlighted above, this suggests that there is valuable latent information in genotypes whose change in abundance does not meet traditional significance cut-offs.

We also trained additional models, where higher-level features (GC content, predicted minimum free energy, base-pairing status at particular rungs of the helix, etc.) were explicitly included. Inclusion did not improve predictive performance, suggesting that emergent properties are captured by models based solely on nucleotide identity at the eight sites. We found that, while inclusion of higher-order features is tempting to increase interpretability, this is a double-edged sword: although higher-order features with large gains can help with interpretation, continuous features or features with more categories can in principle provide more explanatory power for a continuous outcome variable than binary features or features with few categories. Consequently, these features may end up "hogging" predictive power, without necessarily providing greater insight. Exclusive use of nucleotide identities at a given site has the advantage of allowing direct comparison of explanatory power between all features in the model.

To calculate the predictive power of the model (prediction accuracy), one would ordinarily predict fold-change values for the test set (the genotypes left out during training of the model) and compare this to the observed changes. When we do so we obtain correlation coefficients $\rho > 0.83$ for both 30˚C and 37˚C data. Note that, in terms of the variance in fitness across genotypes explained by the model, this estimate arguably better approximates the genetic variance ($V_g$) rather than total phenotypic variance ($V_p = V_g + V_e$). This is because computing fold-changes across several replicates should reduce the environmental part of the variance ($V_e$). To be more conservative, we also calculated fold-changes from five of the six replicates, trained the model on those fold-changes and then tested model performance on the nominal fold-change of the remaining replicate. As expected–given that a single-replicate estimate is bound to be noisier than cross-replicate estimates, correlation coefficients here drop slightly, to $\rho > 0.63$ for both 30˚C and 37˚C.

## Molecular dynamics simulations

The starting structure for simulations was constructed by templating the sequence of the P1/P1ex region of Tet-119(C20A) onto a previously solved P1/P1ex NMR structure (PDB 1HLX) [36]. We then constructed 16 models comprising every single and double base mutation at nucleotides $N_2$ and $N_{21}$. All models were parameterized using the Amber RNA OL3 potentials

for RNA [54], solvated with 14 Å of TIP3P water and neutralized with NaCl. Energy minimization was performed for 2000 steps using combined steepest descent and conjugate gradient methods. Following minimization, 20 ps of classical molecular dynamics (cMD) was performed in the NVT ensemble using a Langevin thermostat [55] to regulate the temperature as we heated up from 0 to 300 K. Following the heat-up phase, we performed 100 ns of cMD in the isobaric/isothermal (NPT) ensemble using the Berendsen barostat [56] to maintain constant pressure during the simulation. All simulations were performed using GPU (CUDA) Version 18.0.0 of PMEMD [57–59] with long-range electrostatic forces treated with Particle-Mesh Ewald summation. RNA base pair properties were calculated using CPPTRAJ [60] and visualized using VMD [61].

## Computation of helix stabilities

Helical stability for P1ex and P10 across all possible $4^8$ and $4^4$ genotypes, respectively, was computed from primary sequence using the efn2 function in RNAStructure v6.2 (https://rna.urmc.rochester.edu/RNAstructure.html). The same bracket notation string was provided for all genotypes so as to force P1ex or P10 pairing as observed in Fig 1A. For simplicity, paired nucleotides were separated in each case by a string of six undefined nucleotides, i.e. bracket notation in all cases was ((((.....)))) for P1ex and ((((((.....)))))) for P10. Forced pairing can lead to very high energy values, which are unlikely to be meaningful as these pairs would not form in practice. We therefore represent energy values in Fig 4C as ordered ranks rather than quantitative values.

## Identification and alignment of Tetrahymena introns

Self-splicing Tetrahymena introns were identified with BLAST (blastn, default parameters, against the nr database), using the sequence of *T. thermophila* ATCC 30382 as bait, and aligned using MAFFT (mafft-linsi—maxiterate 1000).

## Supporting information

**S1 Fig. Fitness and growth data.** (A) Correlation of fitness estimates derived from the DiMSum pipeline and using the DESeq2 framework. (B) Biased distribution of read counts prior to and after selection. As a consequence of library generation, genotypes closer to the master sequence are, on average, more common even prior to selection. (C) Relationship between fitness measured in the pooled-genotype selection experiments, as described in the main text, and doubling time of individual genotypes grown in isolation under selective (+kan, black) and non-selective (-kan, grey) conditions. Doubling time is the median across six biological replicates.
(EPS)

**S2 Fig. The effect of intron insertion into *knt* on colony formation in *E. coli*.**
(EPS)

**S3 Fig. Relative fitness of the Tet-119 genotype and previously described single-mutation derivatives, including our master sequence Tet-119(C20A).** All mutants have previously been shown to have higher splicing activity than Tet-119, including our master sequence Tet-119(C20A), and all exhibit higher fitness in our assay.
(EPS)

**S4 Fig. Correlation of fitness effects at 37˚C and 30˚C.**
(EPS)

**S5 Fig. The native intron in evolutionary context.** (A) The sequence and secondary structure of P1 and P1ex in the native *Tetrahymena thermophila* group I intron and its pre-rRNA environment. Note the differences in $N_2..N_5$ and $N_{18}..N_{21}$ as well as the intervening loop and the downstream exonic sequence compared to the master sequence as displayed in Fig 1A. (B) Excerpt from an alignment of 56 Tetrahymena self-splicing introns, covering P1ex and the adjoining P1 nucleotides. Note that the majority of these sequences were amplified using primers targeting the sequence directly upstream of $N_2$ so explicit nucleotide information for this region is not available.
(EPS)

**S6 Fig. The effect of base-pairing on fitness at different positions.** Fitness is binned according to the types of on-target base-pairing interactions that can be formed by $N_2$-$N_{21}$, $N_3$-$N_{20}$, $N_4$-$N_{19}$ and $N_5$-$N_{18}$ at 30˚C.
(EPS)

**S7 Fig. Stretch and stagger measured at the splice site ($U_1$-$G_{22}$) for all possible nucleotide combinations at $N_2$/$N_{21}$.**
(EPS)

**S8 Fig. Generation of mutant library using site-saturation mutagenesis via two-step PCR.**
(EPS)

**S1 Table. Fitness and read count data for all genotypes.**
(XLSX)

**S2 Table. XGBoost models.**
(DOCX)

**S3 Table. Primers used in this study.**
(DOCX)

**S1 Movie. Molecular dynamics simulation of the $C_2$/$C_{21}$ genotype.** The simulation highlights deformation of minor groove geometry as the splice site U1 rotates out of the helix core and $G_{22}$ mis-pairs with $C_2$.
(MOV)

## Acknowledgments

We thank Feng Guo (UCLA) for the gift of Tet-119(C20A) and mutant plasmids, the LMS Genomics facility for library construction and sequencing, members of the Molecular Systems lab and Romain Strock for discussion, and Ben Lehner, Peter Sarkies, and Karen Sarkisyan for comments on the manuscript.

## Author Contributions

**Conceptualization:** Tobias Warnecke.

**Data curation:** Valerie W. C. Soo, Tobias Warnecke.

**Formal analysis:** Valerie W. C. Soo, Jacob B. Swadling, Andre J. Faure, Tobias Warnecke.

**Funding acquisition:** Valerie W. C. Soo, Tobias Warnecke.

**Investigation:** Valerie W. C. Soo, Jacob B. Swadling, Andre J. Faure, Tobias Warnecke.

**Methodology:** Valerie W. C. Soo, Jacob B. Swadling, Tobias Warnecke.

**Resources:** Andre J. Faure.

**Supervision:** Tobias Warnecke.

**Validation:** Valerie W. C. Soo, Andre J. Faure.

**Visualization:** Jacob B. Swadling, Tobias Warnecke.

**Writing – original draft:** Valerie W. C. Soo, Jacob B. Swadling, Tobias Warnecke.

**Writing – review & editing:** Valerie W. C. Soo, Jacob B. Swadling, Andre J. Faure, Tobias Warnecke.

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
