## [Decision Letter · Decision Letter 0]

25 Nov 2020

Dear Dr Warnecke,

Thank you very much for submitting your Research Article entitled 'Fitness landscape of a dynamic RNA structure' to PLOS Genetics. Your manuscript together with your point to point responses to two reviewers, was fully evaluated at the editorial level and by one independent additional peer reviewer. Based on these evaluations, we concluded that additional wet lab validation experiments are not necessary. However, we ask you to modify the manuscript according to all of your proposed modifications and to all additional reviewer recommendations before we can consider your manuscript for acceptance.

[LINK]

Yours sincerely,

Ivan Matic

Associate Editor

PLOS Genetics

Gregory P. Copenhaver

Editor-in-Chief

PLOS Genetics

Reviewer's Responses to Questions

**Comments to the Authors:**

Reviewer #1: Comments are uploaded as an attachment.

**Have all data underlying the figures and results presented in the manuscript been provided?**

Reviewer #1: Yes

PLOS authors have the option to publish the peer review history of their article (what does this mean?). If published, this will include your full peer review and any attached files.

Reviewer #1: No

---

## [Editor Report · Decision Letter 1]

12 Jan 2021

Dear Dr Warnecke,

We are pleased to inform you that your manuscript entitled "Fitness landscape of a dynamic RNA structure" has been editorially accepted for publication in PLOS Genetics. Congratulations!

Yours sincerely,

Ivan Matic

Associate Editor

PLOS Genetics

Gregory P. Copenhaver

Editor-in-Chief

PLOS Genetics

Comments from the reviewers (if applicable):

**Data Deposition**

http://datadryad.org/submit?journalID=pgenetics&manu=PGENETICS-D-20-01653R1

**Press Queries**

---

## [Editor Report · Acceptance letter]

23 Jan 2021

PGENETICS-D-20-01653R1 

Fitness landscape of a dynamic RNA structure 

Dear Dr Warnecke, 

We are pleased to inform you that your manuscript entitled "Fitness landscape of a dynamic RNA structure" has been formally accepted for publication in PLOS Genetics! Your manuscript is now with our production department and you will be notified of the publication date in due course.

With kind regards,

Alice Ellingham

PLOS Genetics

On behalf of:
